# Study on the Layout of Public Space in Multistory Settlements Based on Outdoor Thermal Environment in Hot-Summer and Cold-Winter Regions of China

Qian Ma [1,2], Lei Shi [1,2], Jiaqi Shi [1,3,*], Simian Liu [1,2], Mengjia Chen [1,2] and Fupeng Zhang [1,2]

1    School of Architecture and Art, Central South University, Changsha 410075, China;
     201311004@csu.edu.cn (Q.M.); shilei@csu.edu.cn (L.S.); 217034@csu.edu.cn (S.L.);
     171301011@csu.edu.cn (M.C.); 201301004@csu.edu.cn (F.Z.)
2    Health Building Research Center, Central South University, Changsha 410075, China
3    College of Architecture, Changsha University of Science & Technology, Changsha 410076, China
*    Correspondence: sjq0831@foxmail.com

**Abstract:** Residential public spaces are closely intertwined with residents' lives as the outdoor thermal environment significantly influences the comfort and safety of outdoor activities. However, in modern designs, factors such as forms, aesthetics and functionalities often take precedence, resulting in the neglect of the microclimate of the settlement's public spaces. In this paper, we established a workflow of "parametric simulation-performance simulation-genetic optimization". By employing the octopus genetic algorithm tool, we conducted experiments on a typical model and set objectives to optimize the winter sunshine duration as well as the thermal comfort during the summer and winter. The results indicated that the average value of the UTCI was optimized for both the summer and winter. This study concludes that altering the layout of public spaces is beneficial for the outdoor microclimate. Additionally, the presence of evenly distributed open node spaces throughout the settlement can improve ventilation in all areas while also protecting it against the winter cold and the dissipation of summer heat. Moreover, it is advisable to position larger public spaces, such as plazas, in the south or southeast. The number of public spaces should gradually decrease in size from the southeast to northwest as this prevents excessive cold winds from traversing in the settlement during the winter.

**Keywords:** residential public space; microclimate; performance-driven optimization; genetic algorithm

## 1. Introduction

With accelerating urbanization, the demand for housing has continued to surge, resulting in a continuous increase in residential land and the development of high-density and high-level residential spaces [1,2]. However, as these areas are no longer able to meet the needs of high-quality development in the new era, some urban construction plans have proposed the development of multistory housing in China. Public spaces are considered a vital component in residential areas and are closely linked to the daily lives of the residents [3]. They provide spaces for rest, socializing and recreational activities. The "Interaction and Space" concept promotes the use of extensive lawns in modern urban areas as they encourage interactions among residents and help to enrich their social lives. Perspective views on public spaces showcase a vibrant variety of activities in these areas [4].

The design of modern public spaces in settlements often prioritizes form, aesthetics or function over the quality of the outdoor thermal environment [5]. Unfortunately, this approach often overlooks the importance of considering the outdoor microclimate and results in the deterioration of the public space environment. To address this issue, researchers have focused on microclimate optimization design as a way to optimize public spaces in settlements. By doing so, we can mitigate the conflict between people and the environment, resulting in a more pleasant and energy-efficient living environment [6,7].

When simulating microclimatic environments, the relationships between buildings and their surroundings, as well as between the buildings themselves, are also taken into consideration. This involves optimizing the overall layout and form of the buildings, including the cluster layout, orientation, streets and open spaces between the buildings. Golony G.S proposed that the influence of urban forms on the urban microclimate should be paid attention to when designing public spaces in 1996 [8]. As early as the 1870s, some scholars had already conducted research on the influence of mesoscale urban forms on the thermal environment, having focused more on neighborhood-scale morphological indicators early on and later gradually carrying out research on specific spaces such as green spaces and square layouts. C.S.B Grimmond conducted a study on the urban isthmus [9] and found that building height, street height-to-width ratio and wall material all had an impact on the OHM (Objective Hysteresis Model) parameters in the OHM. Akira et al. investigated the impact of the building layout and morphological indicators on the thermal environment of the largest residential project in Japan, Domus City [10], and summarized six layout patterns, including the row, enclosure, point, rotating row and diagonal. F Bourbia and H.B Awbi studied the effect of different street aspect ratios on solar radiation [11] and came up with the most suitable aspect ratio for different orientations. A. Yezioro et al. [12] studied the thermal environment of urban plazas with the SHADING model to investigate the relationship between the aspect ratio, surrounding building heights and thermal environment of urban squares, and the results showed that the thermal environment of rectangular squares elongated along the N-S direction was optimal, that of the rectangular squares elongated along the E-W direction was the worst, that of the rectangular squares elongated in the NW-SE and NE-SW directions performed better and the surrounding building heights should not exceed half of the width of the square. Chirag and A. Ramachandraiah [13] studied the effect of the street aspect ratio and greenery coverage on the thermal environment through actual measurements and an ANOVA and found that the correlation between the street aspect ratio or greenery coverage and PET (physiological equivalent temperature) was small, so they proposed a new index called the HXG (read as H cross G) scale that had a satisfactory correlation ($R^2 = 0.648$), showed the product of the aspect ratio and greenery coverage and verified that the HXG had a stronger correlation with thermal comfort. iMohammad et al. [14], by simulating the thermal environment of three different urban forms: the point, row and courtyard, found that the average radiation temperature and direct solar radiation had a greater impact on the thermal environment in different layouts and that the courtyard was the optimal layout for a Dutch thermal environment in June. iGamero et al. [15] studied the thermal environment of high-density slab outdoor spaces by measuring a total of 63 semioutdoor spaces and classifying them into five types: the perimeter buffer (PB), sky terrace (ST), horizontal breezeway (HB), breezeway (BAT) and vertical breezeway (VB). These five types of semioutdoor spaces were then compared in terms of thermal comfort (based on the PMV *), environmental parameters and building form indicators, and it was found that thermal comfort was most desirable factor; the type VB and HB spaces had high comfort, and the lowest comfort was found in the type PB, BAT and ST spaces. In 2023, Dayi [16] simulated 17 tree layout abstraction models by using numerical simulations and found that the correlation between the tree layout and mean radiation temperature and physiological equivalent temperature was very high, with a maximum reduction of 20 °C and 11 °C, respectively. Moreover, the tree layout mainly affects the wind environment, so when considering summer thermal comfort, it is recommended to place trees downstream of the wind. The study of the microclimate in settlements has primarily focused on the morphological indicators of buildings and urban design strategies. However, there is still a gap in the research on the direct impact that outdoor public spaces have on people's comfort during outdoor activities. In this regard, this research takes a unique approach by considering public space in settlements as the main focus of investigation. By incorporating public spaces into the analysis, this research aims to fill the gap in the research on outdoor

comfort in settlements. This approach has the potential to provide valuable insights into the design of public spaces that enhance the overall comfort and livability of settlements.

As technology advances, research on the physics of buildings is increasingly relying on computer simulations rather than traditional field measurements. Artificial intelligence is also becoming more integrated into the field, promising greater efficiency and more comprehensive simulations under various conditions. This shift from field measurements to simulations has opened up new possibilities that were previously not attainable. Specifically, computer simulations allow researchers to examine the complex relationship between urban forms and the microclimate. By modeling different building materials, layouts and site orientations, researchers can better understand how the microclimate is impacted by urban forms. As a result, this approach allows for more in-depth analyses and more opportunities to improve the comfort and livability of urban areas. In 1983, British meteorologist Luke Howard conducted the earliest measurements of the urban heat island effect and proposed the concept in his book "The Climate of London" [17]. Following this, many researchers have studied the urban microclimate through field measurements. Hoyano [18] studied the impact of green space types on the thermal environment through field measurements, and Koichi Nagara [19] studied the greater susceptibility to discomfort near intersections by using both field measurements and subjective evaluation questionnaires. It was not until the development of computer science made it possible to simulate the physical urban environment with software that researchers began to use simulation software to carry out studies. Michael [20] and others used ENVI-met software to simulate the effects of greenery, buildings and street gaps on the thermal environment. Fazia [21] studied the impact of street canyon geometry and sky view factors on the thermal environment by conducting numerical simulations. The application of genetic optimization algorithms allowed the researcher to continually innovate the research methodology. The application of genetic optimization algorithms has led researchers to innovate their methods. Xing Shi [22] integrated Energy Plus into an optimization tool by writing a DOS file and applied a multiobjective genetic algorithm to find the optimal set of Pareto solutions to obtain the best design strategy for office buildings, with the goal of ensuring thermal insulation and energy saving; today, Ecotect [23], Energy Plus [24] and other performance simulation platforms have been widely popularized. However, the performance is only used as a reference condition in the design and cannot be used directly to guide the creation of buildings, and the mutual limitations of the different technology platforms result in a lack of integrated involvement of the environmental influences in the design. As technology evolves and architects master interdisciplinary software, the performance-driven optimization of design is becoming the focus of the moment. Ibrahim [25] used the ladybug tool to study the relationship between three urban form parameters—the courtyard, row and point—and outdoor thermal comfort and energy efficiency, and the results showed a strong correlation between the form parameters and combined the thermal comfort and energy performance, with the building density having the greatest impact on thermal comfort and energy consumption. Yan Hainan [26] constructed a workflow for the rapid prediction and evaluation of the comprehensive performance of office buildings, which was used to predict and evaluate solar radiation, indoor and outdoor thermal comfort and indoor lighting, and it achieved a classification prediction accuracy of 0.77 with a recall of 0.59 and an F-1 of 0.75 through XGBoost, which helps to optimize the comprehensive performance of office buildings in the early design stage. This paper aims to focus on the automatic optimization search of public spaces by using genetic algorithms and optimizing the physical performance of buildings. Previous research has already investigated the commonly used layout methods and drawn valuable conclusions; hence, this study will not delve into creating public spaces with varying layout methods. The objective of this optimization search is to improve microclimate comfort during the summer and winter. This research was conducted by using Rhino(RH) and Grasshopper(GH) software and platforms.

The main research components include:

1.  Conducting a comprehensive review of the existing literature on microclimate research in settlements as well as the research methods that combine building planning and design with environmental physical performance optimization.
2.  Analyzing meteorological data to summarize the basic characteristics of Changsha's climate and creating an abstract model of multistory settlements in Changsha by conducting field research and utilizing web data.
3.  Setting the location and number of public spaces as the independent variables and measuring the dependent variables, which include the average outdoor thermal comfort and sunshine hours in the summer and winter, by conducting a multiobjective search for optimization and utilizing the ideal model as the research object.
4.  Conducting qualitative and quantitative analyses to rigorously analyze the results of the search. The qualitative analysis primarily focuses on the overall layout of the public space, while the quantitative analysis scrutinizes the morphological elements of the public space.

## 2. Methodology

### 2.1. Overview Workflow

Figure 1 shows the framework for this study, which is divided into three parts. The first stage is parametric modeling [27], where an idealized model [28] is abstracted from the research case and modeled in RH in a parametric way. Then comes the performance simulation part, where the climate conditions and site surroundings are set by using the ladybug (LB) tool to evaluate the performance of the outdoor physical environment of the building by considering the solar radiation, wind environment and thermal environment. The third part is the evaluation and optimization [29], where the location and area of the public spaces are set as the independent variables, attaining optimal outdoor comfort is the objective and the multiobjective octopus genetic optimization tool is used to analyze the impact of the changes in the layout of the public spaces on the microclimate of the settlement. The tools and methods used in each part will be described in detail in the subsequent sections.

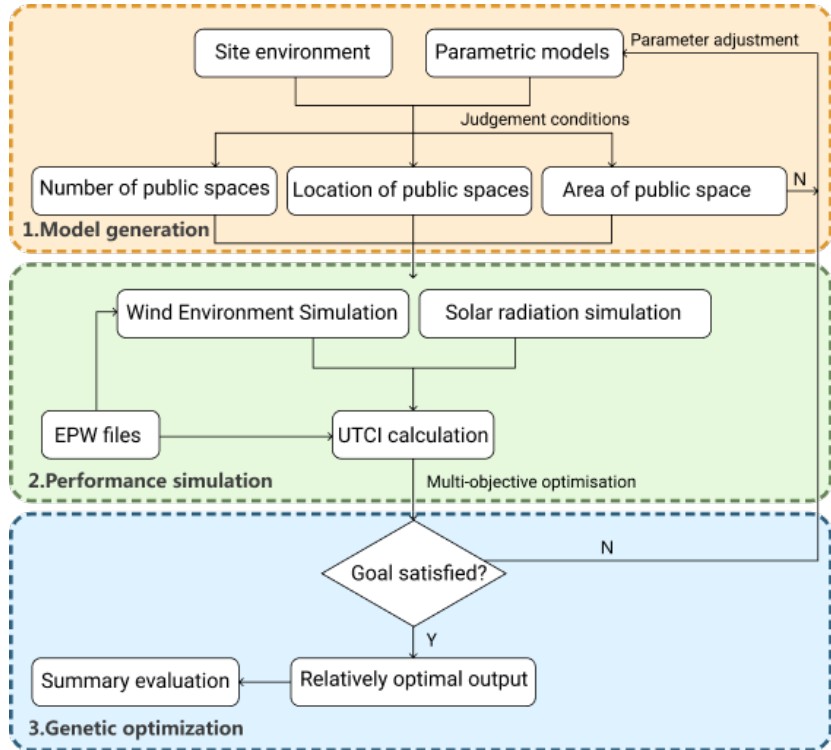

**Figure 1.** Experimental method framework.

## 2.2. Case Study

### 2.2.1. Study Area

This study focused on the hot summer and cold winter region [30], which is known for its severe climate discomfort in the summer and winter [31]. This region covers a vast area of China, including 14 provinces and one fifth of the country's total land area (Figure 2). The climate is characterized by high temperature and humidity in the summer; wetness and coldness in the winter; a small daily temperature difference; high annual precipitation, with a rainy season in the late summer and early spring; and low sunshine [32]. Therefore, it is of theoretical and practical significance to study the layout [33] design of the public spaces in this region with the physical performance of the built environment as the guide. Changsha [34] is a typical representative city of the hot summer and cold winter region. The demand for cooling and heating in residential areas of the city is high throughout the year, which results in a great drain on energy consumption. Therefore, microclimate-oriented settlement design is not only conducive to increasing the level of outdoor comfort, but also to reducing energy consumption [35].

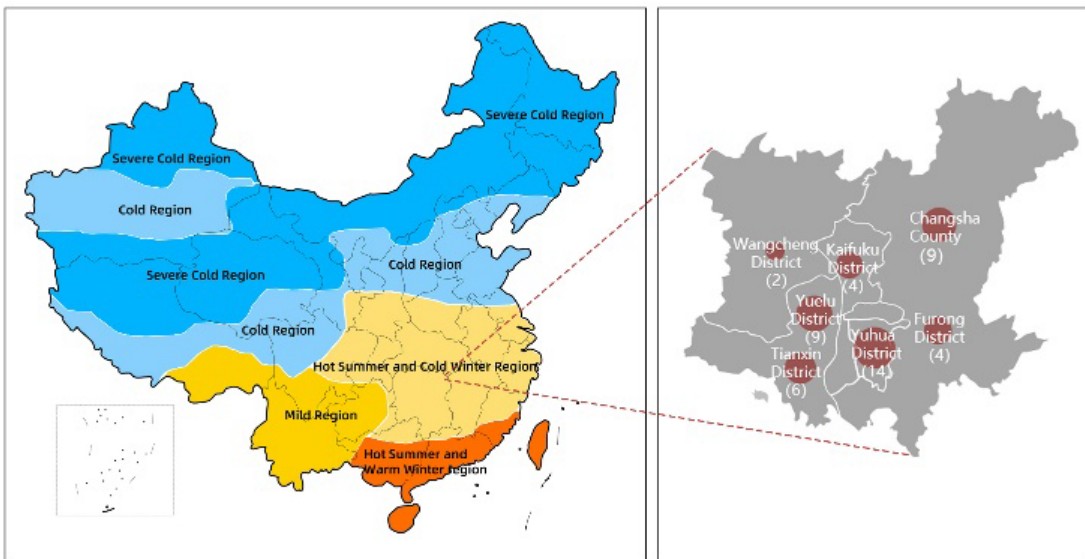

**Figure 2.** Hot summer and cold winter region location and distribution of 48 sample settlements in Changsha.

Residential buildings [36] exhibit a repetitive architectural form and are typically arranged in such a way that a residential area can be considered a basic unit. The microclimate of this basic unit can, to a certain extent, reflect the overall microclimate of the residential area. Unlike other types of buildings in a city, the design and placement of public spaces within a residential area can significantly impact the physical environment around the building. Factors such as the location, size and layout of these public spaces all play a crucial role in shaping the microclimate of the residential area. Therefore, optimizing the design and layout of public spaces within a residential area is essential to mitigate energy consumption and improve outdoor comfort in the built environment [37].

### 2.2.2. Selection of Sample Residential Clusters

The research object of this paper is the layout of public spaces in a multistory residential area, so the research sample should meet the following characteristics: 1. The residential area contains at least one residential group, and the number of buildings is not less than 6; 2. According to the General Principles of Civil Building Design, a multistory residential building is 4–6 stories, so the sample is a pure multistory district or a mixed district with mainly multistory buildings; and 3. In the literature review, it was found that the commercial houses in Changsha were mainly built after 2000, so the construction time will

be taken into account when selecting the study sample. By means of arcgis data crawling and a field survey, 48 multistory residential districts in seven districts of Changsha were selected as the study samples, and the distribution of these 48 samples is shown in Figure 2. The six districts are the Furong District, Tianxin District, Yuelu District, Kaifu District, Yuhua District and Wangcheng District. Through a mathematical statistical analysis, the basic data of these 48 samples were compiled in terms of the plot size, building density, floor area ratio, building orientation and layout characteristics. The selection of each morphological [38] layout indicator should also be calculated according to the theory of normal distribution. The data collected for this study were first normalized by using origin data-processing software, and a normalization curve was used to determine the typical values of the residential indicators.

*2.3. Model Generation*

2.3.1. Field Measurements

Before creating the ideal model for the residential community, it is essential to select the appropriate design indexes for the community's form. In the scheme design stage, the design indexes mainly include the building density, green area ratio, plot ratio, land scale, building scale, building layout, average building height, building orientation and building spacing. It is important to follow two principles when selecting these settlement design indicators: (1) the indicators should be the design elements of multistory settlements, and (2) the indicators must impact the layout of the public spaces in settlements.

Based on the above principles, the indicators listed in the specification satisfied the requirement. Since the type of green space has a significant impact on the thermal environment of the settlement, the green space rate was not included in the statistics. Additionally, since multistory settlements in Changsha typically consist of six-story houses, the volume ratio and average building height were not considered. Therefore, the residential indicators considered in this study were: (1) land size; (2) building density; (3) building scale; (4) building orientation; and (5) building spacing (Figure 3).

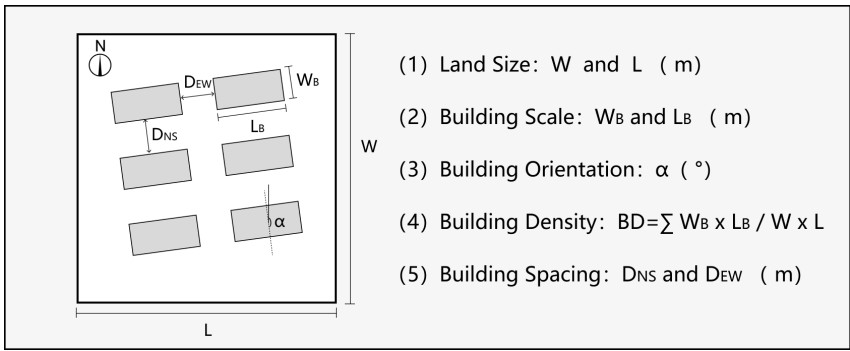

**Figure 3.** Detailed description of settlement indicators.

The distribution of the statistical data on the residential indicators was similar to a normal distribution, and the data could be processed based on the theory of normal distribution when selecting the typical values for each morphological layout indicator. Specifically, for a set of data, the mean ($\mu$) and standard deviation ($\sigma$) are used to determine a range from $\mu - \sigma$ to $\mu + \sigma$, where the mean of the data within this range is taken as the typical value if the proportion of the data sample exceeds 68.3%. This data treatment is based on the management statistics approach provided by Andrew Siegel in Practical Business Statistics, which reduces the impact of outliers and makes typical values more representative [39]. The calculations were first normalized by using origin data-processing software to determine if the range of $\mu - \sigma$ to $\mu + \sigma$ was greater than 68.2% according to normal distribution theory, and then the mean value of the data within this range was calculated as the typical value (Figures 4–6).

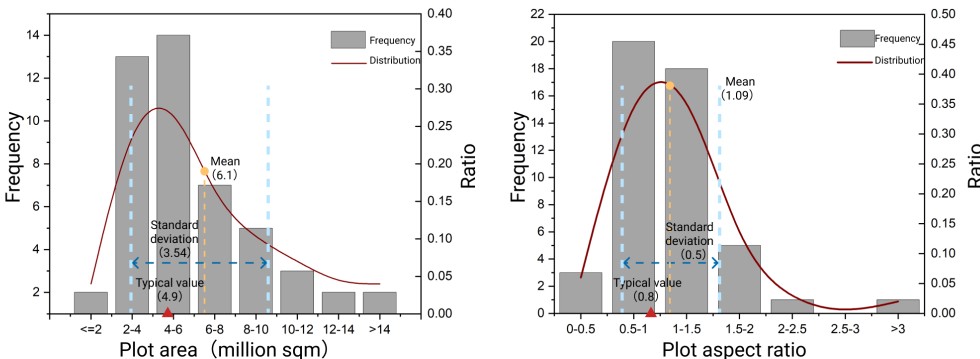

**Figure 4.** Histogram of plot area and plot aspect ratio of 48 multistory settlements in Changsha City.

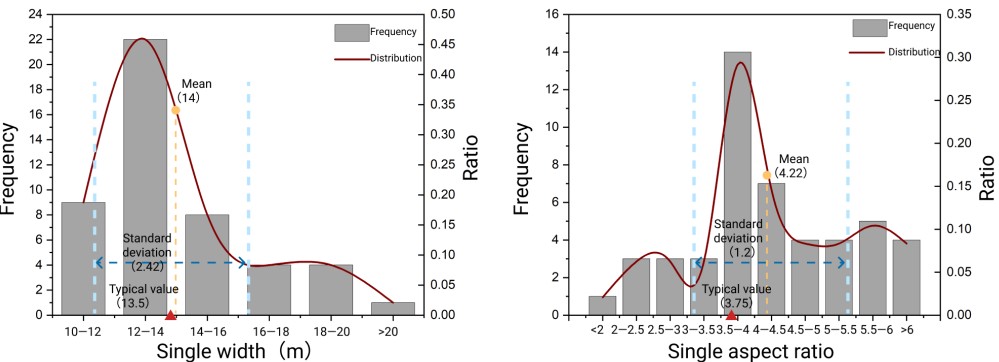

**Figure 5.** Histogram of single width and single aspect ratio of 48 multistory settlements in Changsha City.

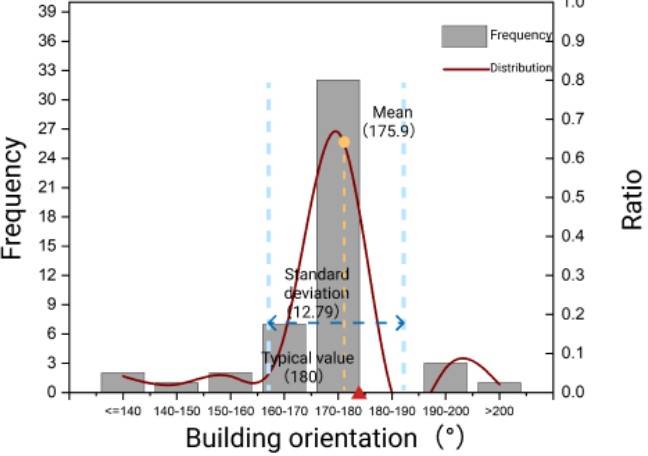

**Figure 6.** Histogram of building orientation of 48 multistory settlements in Changsha City.

### 2.3.2. Typical Models

Based on previous research and the Technical Regulations of Changsha City Planning Management [15], it was determined that the minimum net land area for multistory residential buildings should be 10,000 m², the spacing of buildings in a parallel arrangement should be greater than or equal to 1.1 times the building height while meeting sunlight requirements and the minimum spacing of the building hill wall surface should be 7 m. The ideal model for this project was finalized with a plot size of 250 m × 200 m, a building monolith size of 13.8 m × 52 m, a building spacing of 21 m from north to south and 10 m from east to west, a monolith orientation of 180° and a uniform building height of 6 stories with a floor height of 2.8 m. The finalized model is graphically represented in Figure 7.

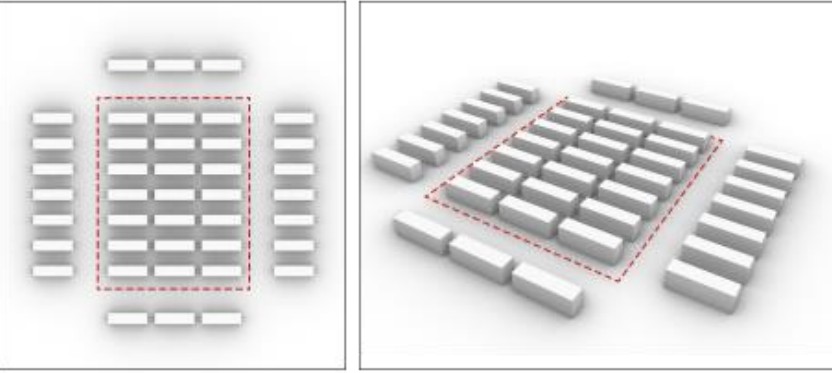

**Figure 7.** Schematic diagram of the ideal model of multistory settlements.

*2.4. Simulation and Optimization*

2.4.1. Computer Simulations

Outdoor thermal comfort evaluation indicators can be divided into three categories. These include thermal risk indicators based on a regression analysis, such as the Wet Bulb Globe Temperature (WBGT) [40]; thermal comfort indicators based on steady-state heat transfer models, such as the physiological equivalent temperature (PET) [41] and physiological subjective temperature (PST) [42]; and thermal comfort indicators based on dynamic heat transfer models, such as the Universal Thermal Climate Index (UTCI) [43]. The thermal comfort index is based on dynamic heat transfer models, such as the Universal Thermal Climate Index (UTCI). The Wet Bulb Black Globe Temperature (WBGT) is often used to evaluate the thermal environment in industrial workplaces and the outdoor areas of buildings. Thermal comfort indicators are based on steady-state heat transfer models and follow the results of indoor thermal comfort research; take into account the deviations from indoor and outdoor conditions; make appropriate corrections to indoor indicators; take into account factors such as body temperature regulation, heat exchange in the human environment and the heat transfer characteristics of the human body during movement; and establish indicators such as the physiological equivalent temperature (PET) and physiological subjective temperature (PST). However, such index evaluation models are based on the assumption that environmental conditions are stable and that the human body is in contact with the environment for a long time and reaches thermal equilibrium. They do not take into account the fact that the actual outdoor environment is constantly changing and that the heat load on the human body is changing all the time [41]. The Universal Thermal Climate Index (UTCI) evaluates the thermal comfort of the human body under outdoor climatic conditions and is similar to the Predicted Mean Vote (PMV), which is used in the evaluation of indoor thermal environments [44] and is based on the iterative calculation of a thermoregulatory model to obtain the dynamic equivalent ambient temperature of a person in a reference environment, which reflects the thermal sensation of the human body [44]. The performance of the UTCI shows that it can be used to evaluate multiclimatic and complex climatic conditions, and it has been compared with several outdoor thermal comfort indices. The UTCI has been shown to be suitable to evaluate multiple and complex climatic conditions and has demonstrated greater accuracy when compared with multiple outdoor thermal comfort indices [43].

As outdoor environments are nonstationary, it is essential to consider the influence of nonphysical factors when selecting indicators. Given that Changsha is located in a region that experiences hot summers and cold winters, microclimate evaluation indicators need to account for both summer and winter thermal comfort. International common thermal climate indicators address physiological dynamics and meteorological data and include values for both summer and winter. Therefore, the Universal Thermal Climate Index (UTCI) is a suitable thermal comfort index to evaluate the outdoor thermal environment explored in this paper (Figure 8).

| UTCI (°C) range | Stress Category |
|---|---|
| above +46 | extreme heat stress |
| +38 to +46 | very strong heat stress |
| +32 to +38 | strong heat stress |
| +26 to +32 | moderate heat stress |
| +9 to +26 | no thermal stress |
| +9 to 0 | slight cold stress |
| 0 to −13 | moderate cold stress |
| −13 to −27 | strong cold stress |
| −27 to −40 | very strong cold stress |
| below −40 | extreme cold stress |

**Figure 8.** UTCI comfort level classification.

In RH and GH, the Honeybee_UTCI Comfort Map [45] can call Open studio to calculate the UTCI values. Firstly, the epw file of the weather in Changsha City is obtained from the US Department of Internal Sources website, and then the data are imported into the Import EPW battery. The period selected for calculation in this paper was 8:00–20:00 for the typical weeks of 22–28 June and 22–28 December for the summer and winter, respectively, and the analysis Pcriod battery was accessed to filter out the time interval required for the calculation; then, the data were filtered out by using the Import EPW battery, which can parse the EPW file to determine the location, dry bulb temperature, humidity, wind speed, wind direction, etc., and import the corresponding values into the Honeybee_UTCI Comfort Map to calculate the UTCI values, where the temperature and humidity, MRT, wind speed, wind direction are imported. The temperature and humidity values are imported here, while the MRT, wind speed, wind direction and other values are simulated by other modules and imported [46].

After importing the 3D model into GH, it is necessary to create the calculation grid and the building surface required for the Honeybee calculation [45] by using the Fcae battery and the Model battery to convert the Brep and surface into a digital model that the Honeybee battery calculation can recognize. The GenPts, Sensor Grid and three batteries of the AssignGridViews can divide the site into n grids, whereby each grid corresponds to the value of the simulation calculation. The _geometry end is connected to the battery surface of the building site and the _grid_size is connected to the value of the grid; the smaller the value, the higher the number of grids, and thus the higher the degree of refinement, according to the performance of the computer. The Sensor Grid can be understood as a sensor, and its _positions end can be connected to the points divided by the GenPts module. The AssignGridViews is used to connect the model and grid and output the results to the UTCI calculation module.

To simulate the wind environment [46], we first converted the model into a butterfly model by using create BF Geometry; then, we created an analysis case by using create case from tunnel, and then we entered the case name in the _name field. This name will also automatically generate a case folder. Next, we used the block Mesh cell to mesh the analysis, and the next step was to mesh the analysis object with a block Mesh cell,

and after connecting the analysis surface and point boundaries in the model generation module, the final calculation was performed by accessing the solution. The final result was automatically saved in the folder, and it contained a record of the wind speed values at each point. The real time wind speed from the record file could be called directly when calculating the UTCI.

To simulate the thermal environment, the Honeybee_UTCI Comfort Map battery was used. Its input side requires a connection to the model, an epw file, a ddy file and the calculation time to be set. The advantage of using this battery for calculation is that it has a run_settings_input port, which can call more CPU resources of the computer and improve the calculation efficiency. On the output side, there are the UTCI and env_conds. The UTCI values need to be converted by using the Thermal Mtx battery and then inputted to the visual display battery. The env_conds output port includes values for solar radiation, air temperature and air humidity, and the EnvMtx battery can be connected to select the desired outputs. If the value inputted is "0", the corresponding output is the MRT value [46].

### 2.4.2. Multiple Objective Optimization
Parameter Setting

To meet the building density standard of 30%, it is necessary to add five new node spaces to the existing row layout. As the locations of these node spaces are not predetermined, the gene pool command can be used to randomly delete building blocks. This involves selecting a total of five blocks for deletion, with the existing 21 blocks numbered by using the Area and Points battery as represented in Figure 9. Due to the random selection process, there may be repeated deletions of the same block. The Delete Consecutive module can be used to eliminate these repeated deletions before proceeding to the next step.

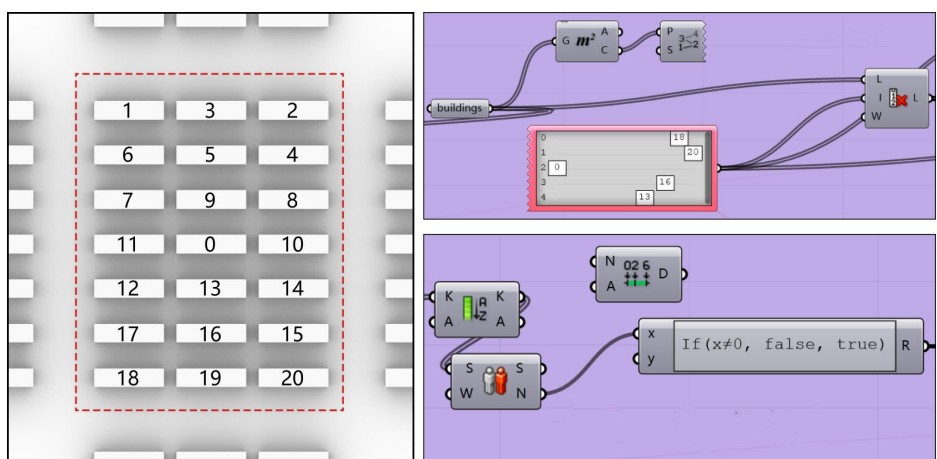

**Figure 9.** Schematic diagram of the setting of morphological variables.

The octopus optimization algorithm tool [36] supports multiobjective optimization, and the climate of Changsha City reflects typical cold winter and hot summer characteristics, so the thermal environment optimization objectives need to be considered for both the summer and winter seasons by taking into account summer heat dissipation and winter warmth. The three optimization objectives under comprehensive consideration are summarized as follows:

(1) Minimize the mean value of the UTCI in the summer;
(2) Maximum the average value of the UTCI in the winter;
(3) Minimize the percentage of days where the sunshine hours are less than 2 h in the winter.

Unlike the Gapagos optimization plug-in in RH and GH, the octopus tool allows for multiobjective optimization, so the parameters of the algorithm need to be set, and the

meaning of each parameter needs to be understood before setting the parameters. The probability, Mutation Rate, Crossover Rate, Population Size, Max. Generations and the specific parameters of the algorithm are shown in Table 1.

**Table 1.** Octopus parameter setting.

| Parameter | Values |
|---|---|
| Variable parameters | Random to cut cell |
| Test grid | 2 m |
| Optimization objective | Summer mean UTCI; winter mean UTCI |
| Elitism | 0.3 |
| Mut. Probability | 0.1 |
| Mutation Rate | 0.6 |
| Crossover Rate | 0.9 |
| Population Size | 50 |
| Max. Generations | 0 |
| Convergence mechanism | Hype Reduction |
| Mutation mechanism | Hype Mutation |

*2.5. Quantification of Public Space*

As the public spaces in settlements have not been systematically studied, there is some difficulty in quantitatively describing public spaces, and quantitative indicators from other disciplines need to be drawn upon. Landscape ecology is also a discipline that studies the relationship between spatial patterns and ecology, where landscape patterns reflect the characteristics of the landscape structure, and there are already many landscape indicators that quantitatively describe the distribution characteristics of the landscape. Public spaces are similar to landscapes only in terms of their spatial morphology, so references to landscape indicators provide new ideas to quantitatively analyze the layout of public spaces. It has also been shown that the location and size of landscape patches have an impact on the microclimate of the location [47]. Based on the research on landscape indicators by various scholars [47,48], the following indicators were selected for the quantitative description of public spaces in settlements.

2.5.1. Location

Dayi Lai [48], when modeling the location of a green space layout, placed different types of green spaces in the southeast, southwest, northeast and northwest corners of the site, and this paper also uses this approach to describe the layout of public spaces. This description is used to define the location with reference to the center of the site, so the layout locations used in this study may be the center, due north, due south, due west, due east, northwest, northeast, southwest and southeast corners. In terms of the internal and external relationships, the locations can be described as being at the edge and interior. The edge refers to the junction between public space and the surrounding environment, such as the adjacent location of public spaces to urban parks, commercial streets, main roads, etc. The interior refers to the position of public spaces within the community, such as public squares, small gardens, community gyms, etc. (Figure 10).

2.5.2. Dispersion

Dispersion [49] refers to the extent to which the distribution of similar indicators deviates from the distribution of central indicators. In this paper, dispersion refers to the degree of dispersion of the distribution of public spaces in a plot. As shown in the figure below, the larger the expansion space, the more discrete the layout in the same total spatial area. In the case of different areas, the area ratio can be used to define the dispersion of the layout. The greater the dispersion, the larger the ratio of the expansion area to the actual public space area, and the smaller the dispersion, the smaller the ratio (Figure 11).

The minimum value of the ratio is 1, which is achieved when all five public spaces are concentrated together. The formula is expressed as follows:

$$D = \frac{S_1}{S_2} \tag{1}$$

Note: D represents the dispersion, $S_1$ represents the total area of public space and $S_2$ represents the area of the public space expansion.

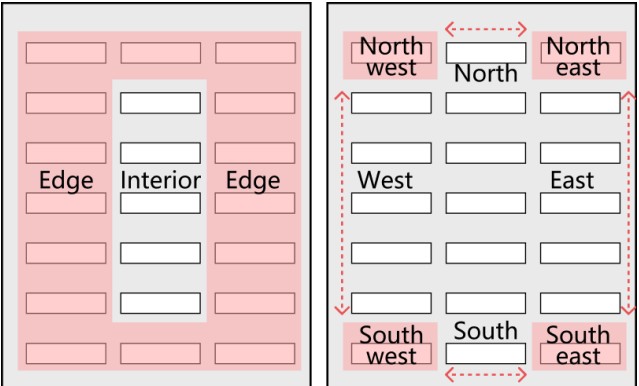

**Figure 10.** Directional indicator.

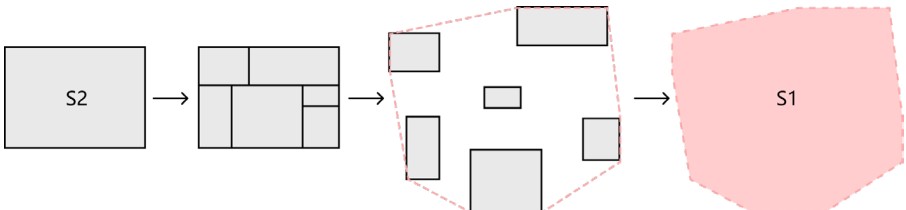

**Figure 11.** Schematic.

### 2.5.3. Upwind Opening Rate

In this paper, the angle between the open space and the wind is considered, and the size of the air inlet is characterized by the upwind opening rate, as shown in Figure 12. The angle with the wind is the clockwise angle between the center of the open space and the wind direction. The formula for calculating the upwind opening rate is as follows:

$$R = 1 - \frac{\sum_{n=1}^{n} a_i}{a} \tag{2}$$

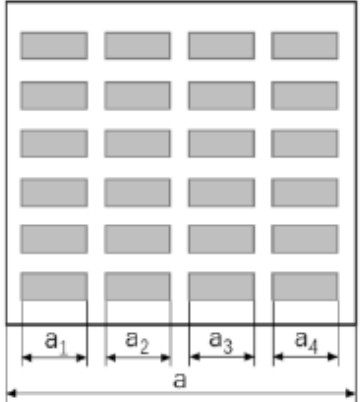

**Figure 12.** Schematic.

Note: R—opening rate upwind, %;

$a_i$—Projected length of the first row of buildings on the upwind side of the site in the wind direction, m;

$a$—Projected length of the site in the wind direction, m.

## 3. Results

### 3.1. Overall Optimization Results

In this experiment, based on previous climate parameters, a genetic algorithm [50] was carried out to find the optimum public space layout, whereby five node-type public space locations were used as the independent variables and optimizing the mean value of the UTCI in the summer and winter was the objective, which took a total of 52 h. The mean values of the summer UTCI and winter UTCI converged, but the percentage of days where the sunshine hours were less than two fluctuated during the two seasons without obvious signs of convergence, indicating that there were checks and balances among the three optimization objectives and the correlation between sunshine hours and the summer and winter UTCI was weak when changing the layout of the public spaces, so the subsequent discussion focuses more on the thermal environment in the summer and winter (Figures 13–15). The three objective values fluctuated up and down during the genetic optimization search due to the genetic algorithms causing continuous fluctuations in the objective values, which occurred so that the algorithm could explore all solutions, the premature convergence of the values could be prevented and a local optimum solution could be obtained [26].

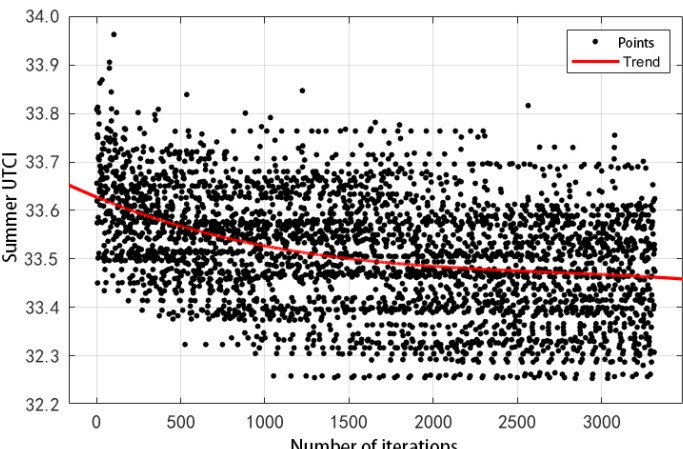

**Figure 13.** Summer UTCI optimization process.

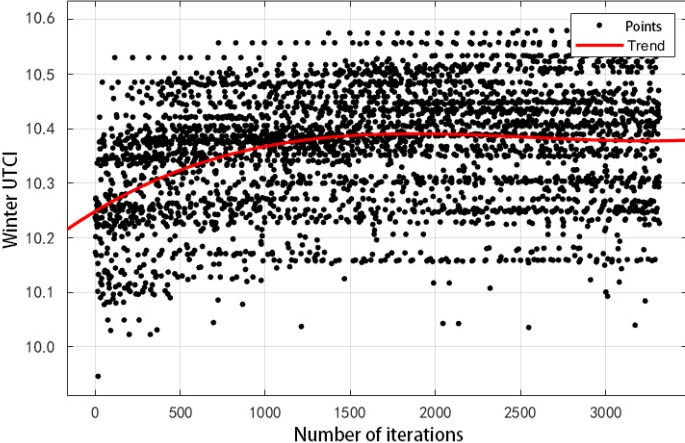

**Figure 14.** Winter UTCI optimization process.

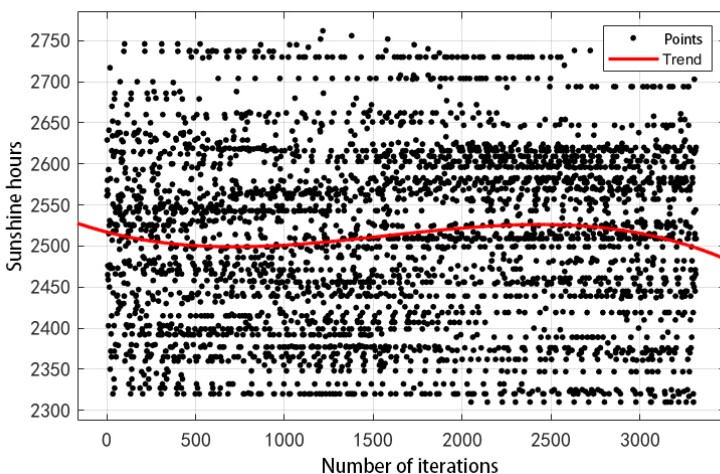

**Figure 15.** Sunshine duration.

During the whole optimization process, two of the three dependent variables reached convergence after the optimization iterations occurred, in which the mean value of the UTCI in the summer decreased from 33.97 °C to 33.26 °C with a difference of 0.71 °C and the mean value of the UTCI in the winter increased from 9.95 °C to 10.58 °C with a total increase of 0.63 °C; because the objective was to optimize the thermal comfort, the optimization objectives were achieved (Figure 16).

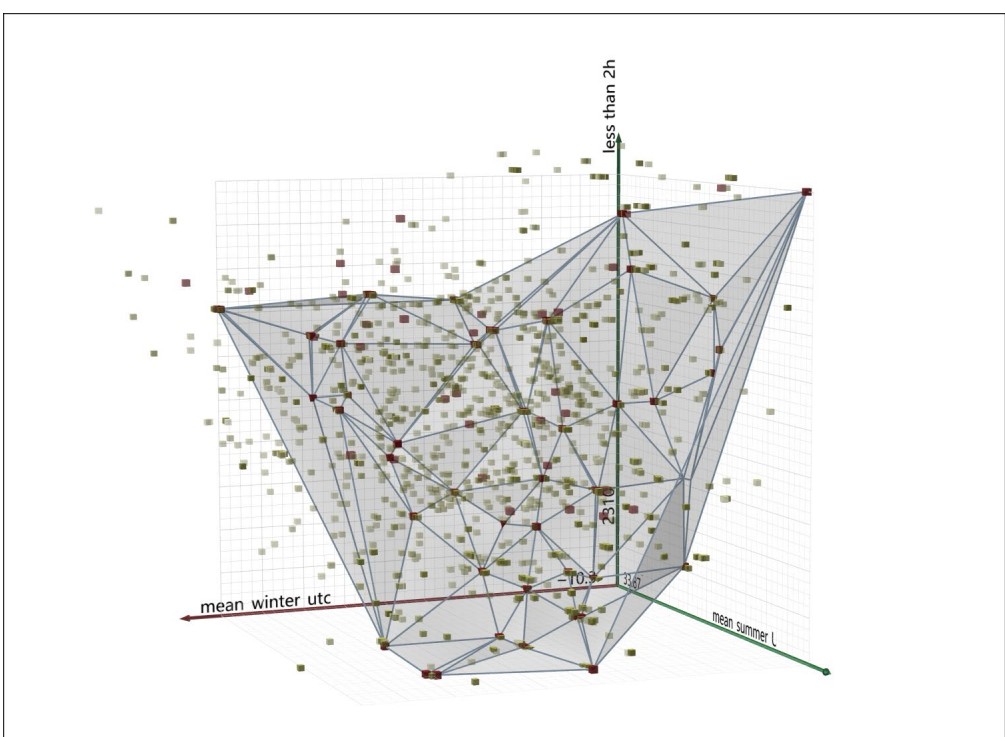

**Figure 16.** Octopus optimization results.

Only the relationship between the mean UTCI values in the summer and winter was considered when analyzing the Pareto evolution [51]. While optimizing the summer and winter UTCI, it was observed that the mean summer UTCI and the mean winter UTCI reached convergence; however, the winter sunshine hours were shown to continuously fluctuate. This implies that the objective of optimizing the winter sunshine hours cannot outweigh the advantages and disadvantages of the other two optimization objectives. To address this issue and make the analysis more accurate, the three-dimensional coordinates

were converted to two-dimensional coordinates, and the optimization of the summer UTCI and winter UTCI were used as the primary objectives for the analysis. Additionally, the Pareto solutions for the 5th, 11th, 18th, 25th, 32nd and 39th generations were plotted with the mean thermal comfort in the summer as the horizontal axis (red axis) and the mean thermal comfort in the winter as the vertical axis (green axis). As shown in Figure 17, the overall trend of the solution distribution increased and then decreased and gradually approached the origin of the coordinates. As the number of generations increased, the solution gradually resulted in the balance of winter comfort and summer comfort.

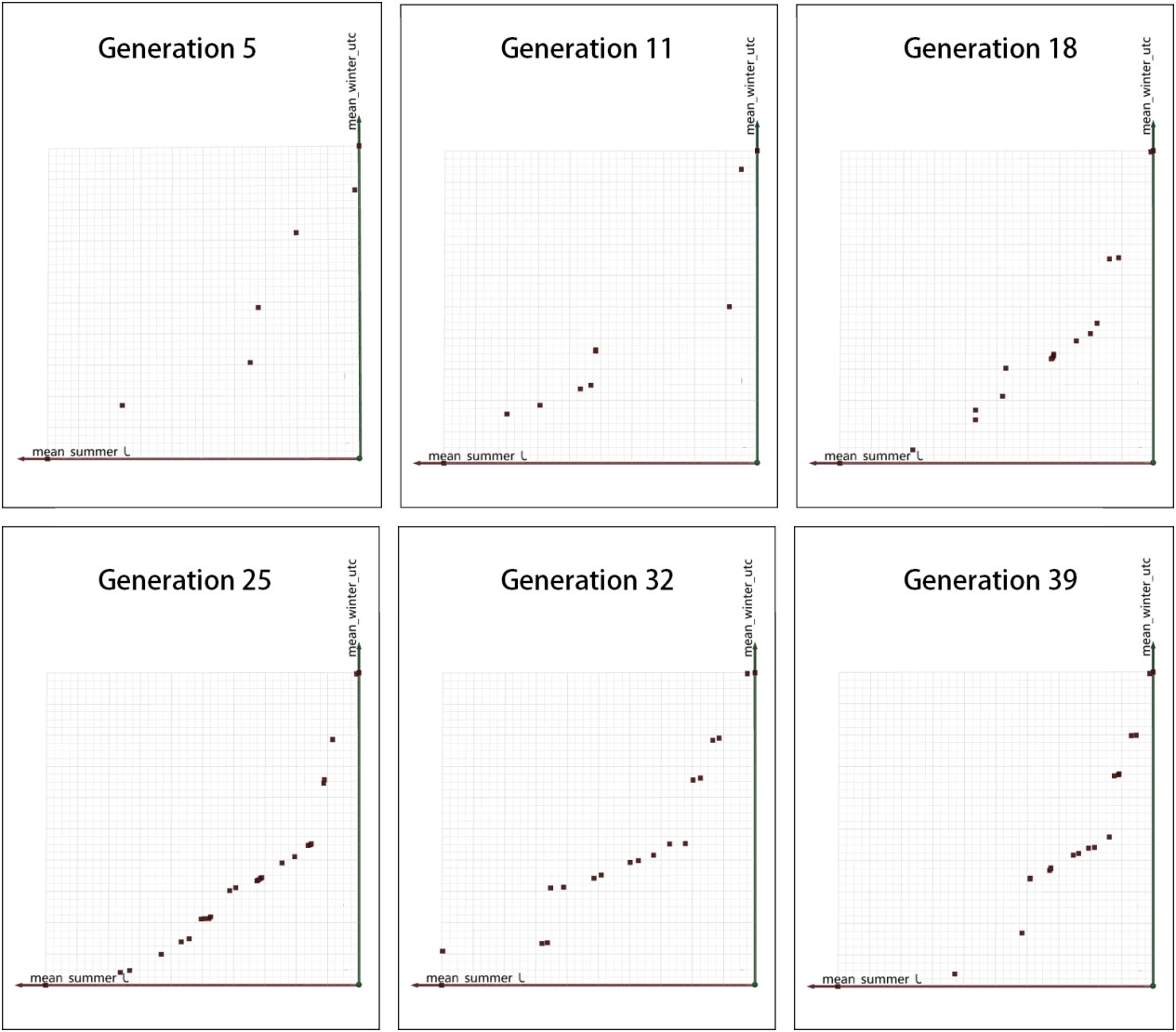

**Figure 17.** Evolution of Pareto solutions by generation.

From the Pareto front solutions generated during the iterative optimization process, we selected 20 cases at random. Throughout the optimization process, random deletions were performed on the blocks. Figure 18 shows that the process of block deletion went from being disorderly to more regular over time. The number of blocks deleted on the west and north sides decreased while those on the east and south sides increased. We also observed a higher frequency of block deletions in the central location during the later stages of the process.

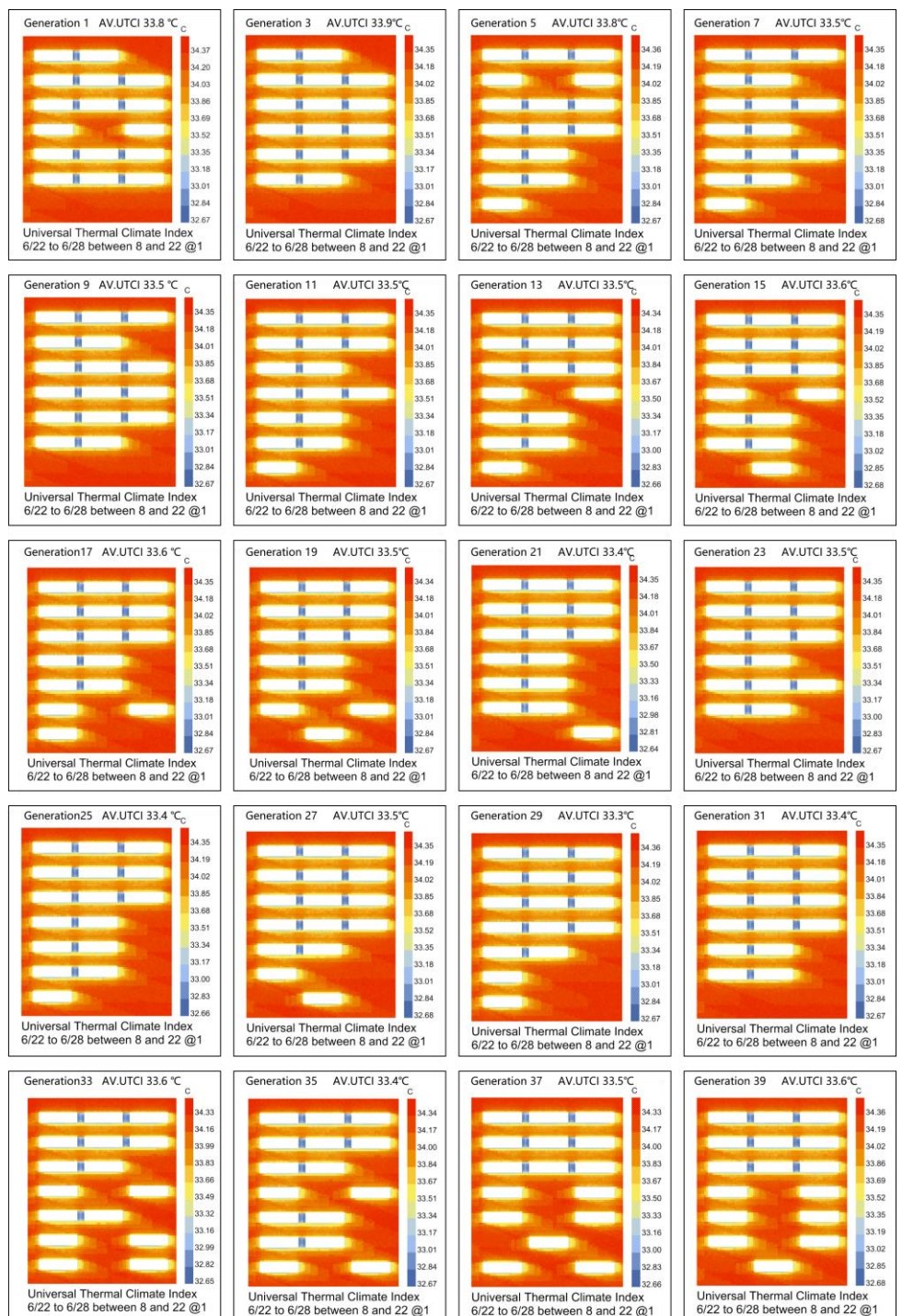

**Figure 18.** Summer UTCI variation chart of optimization process cases.

## 3.2. Location

In the process of multiobjective [52,53] optimization seeking, the positions of the node space appearances gradually converged after being disorderly. After conducting a statistical analysis on the node records that were collecting during the optimization process as shown in Figure 19, we found that the number and probability of each node being selected varied, and the more times the censored number was selected during this experiment, the better the optimization [54] effect was for the overall UTCI value. From the frequency graph of the statistical results, it can be seen that the points with a higher frequency of occurrence were distributed in the southeast corner and middle of the plot, whereby the lowest frequency of

occurrence occurred in the west and north. The body number 19 was selected for censoring the most times at 13.11%, followed by the body numbers 20, 15 and 14; additionally, the body in the center of the plot was also selected for censoring more frequently, and the body point that was selected for censoring the least was point number 6. The difference between the highest and lowest was large, indicating that there was a clear tendency for the genetic algorithm to find the best body numbers.

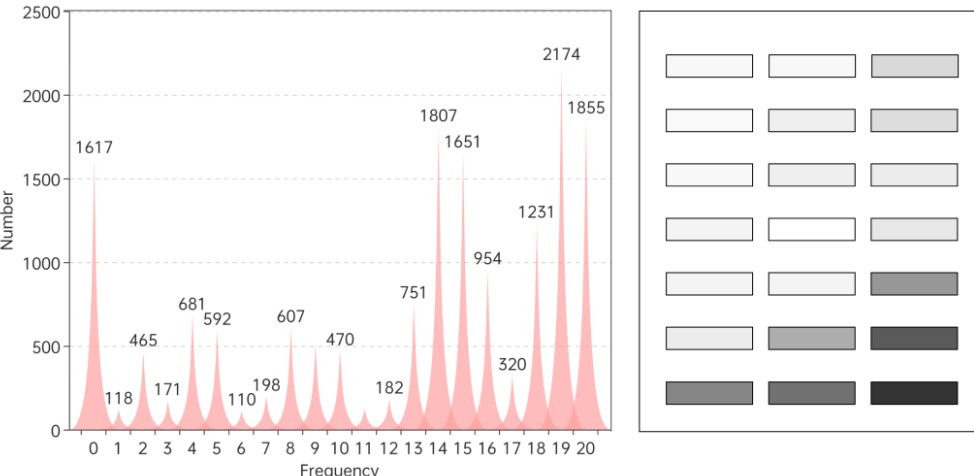

**Figure 19.** Point frequency distribution map of nodal spatial optimization process.

These results suggest that there is a correlation between the geometric properties of public spaces and thermal comfort in the summer and winter. The openness of public spaces is conducive to the flow of prevailing winds, which increases the wind speed and removes the heat in the summer but increases the wind speed and reduces the thermal comfort in the winter. In the summer, when there is a certain number of public spaces, the process of finding the optimum convergence involves a tradeoff between the two. When designing the layout of public spaces in multistory settlements, priority can be given to the south center and the most southerly position on the east side of public spaces when considering heat dissipation in the summer and warmth in the winter, which means that the west and north sides do not need to be treated as public space nodes.

*3.3. Dispersion*

The graphs indicated that as the dispersion of the public spaces increased, the average UTCI tended to rise during the summer and decline during the winter (Figures 20 and 21). This indicates that public spaces that are too scattered in a residential area have a less favorable UTCI in both the summer and winter, while public spaces that are more concentrated or have a larger number of buildings and a more concentrated distribution are more favorable in terms of comfort, so the process of finding the optimal solution involves constantly weighing the needs of public spaces during both the summer and winter to find a better solution set. To provide a further explanation, the dispersion of public spaces refers to how spread out or concentrated they are within a residential area. When public spaces are too scattered, it can lead to less favorable thermal comfort levels for people during both the summer and winter. This is because dispersed public spaces may not provide sufficient shade, ventilation [55,56] or greenery to help cool or warm the surrounding areas, which results in discomfort for those who use them. In contrast, public spaces that are more concentrated or have a larger number of buildings and a more concentrated distribution can create microclimates [57] that are more favorable for thermal comfort during both the summer and winter.

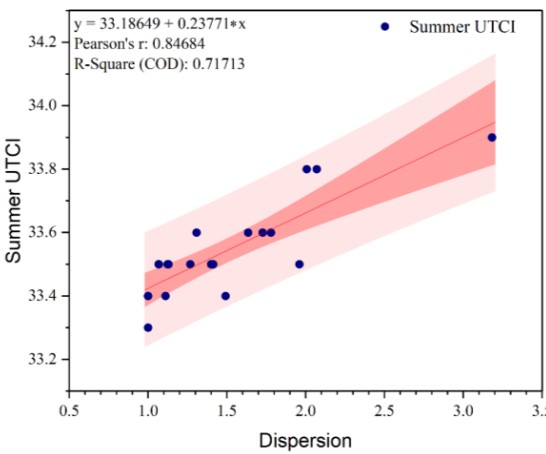

**Figure 20.** Linear fit of dispersion of summer UTCI.

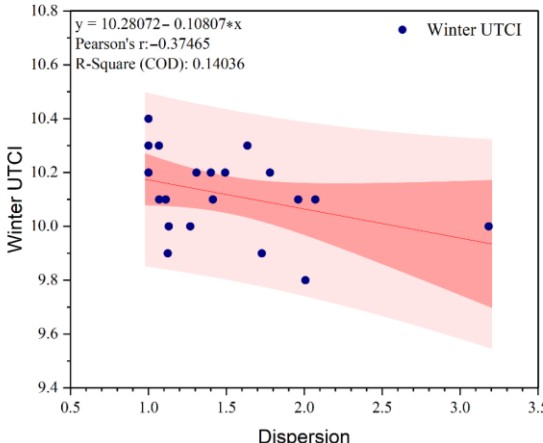

**Figure 21.** Linear fit of dispersion of winter UTCI.

### 3.4. Relationship with the Windward Entrance

The results of the statistical analysis on the upwind opening rate are presented in Table 2. The analysis demonstrated that there was no clear linear connection between the upwind opening rate and the UTCI during both the summer and winter. Additionally, the upwind opening rate was consistently greater than 0.33 in the Pareto front solution over every generation of the optimization process. These findings suggest that when designing a residential area, introducing openings in the first row of buildings can enhance the thermal comfort of the entire zone. By incorporating upwind openings and considering the prevailing wind direction during different seasons, other public space layout configurations can be aligned to facilitate the flow of air within the residential area. This linked approach to public space design can further contribute to enhancing thermal comfort in residential areas.

### 3.5. Larger Squares

During the typical generation of the genetic optimization process, the layout scheme of the larger square [58,59] was screened out, and the screening principle was that the adjacent censored body blocks were greater than or equal to three. Since the total number of censors was five, three censors or more were greater than the average and were considered to reflect a larger common space, and the final screening results are shown in Table 3. As can be seen from the table, 90% of the larger squares were distributed at the edges of the plots, one scheme distributed the public spaces inside the plots [60] and placed the spaces in the center of the plots and 82.4% of the larger squares that were located at the edges were in the southeast end while the rest were distributed in the south and east (Table 4). These results were obtained by using the 3.6. Pareto Decentralization Typical Solution.

**Table 2.** Distribution of upwind opening rates.

| Generations | Winter UTCI | Summer UTCI | Upwind Opening Rates |
|---|---|---|---|
| g1 | 9.8 | 33.8 | 1 |
| g3 | 10 | 33.9 | 1 |
| g5 | 10.1 | 33.8 | 0.74 |
| g7 | 9.9 | 33.5 | 0.74 |
| g9 | 10.2 | 33.5 | 1 |
| g11 | 10.1 | 33.5 | 0.74 |
| g13 | 10.1 | 33.5 | 0.74 |
| g15 | 9.9 | 33.6 | 0.74 |
| g17 | 10.2 | 33.6 | 0.74 |
| g19 | 10 | 33.5 | 0.74 |
| g21 | 10.1 | 33.4 | 0.74 |
| g23 | 10.3 | 33.5 | 1 |
| g25 | 10.3 | 33.4 | 0.74 |
| g27 | 10 | 33.5 | 0.74 |
| g29 | 10.2 | 33.3 | 0.74 |
| g31 | 10.4 | 33.4 | 1 |
| g33 | 10.3 | 33.6 | 0.48 |
| g35 | 10.2 | 33.4 | 0.48 |
| g37 | 10.1 | 33.5 | 0.48 |
| g39 | 10.2 | 33.6 | 0.74 |

**Table 3.** Distribution of larger squares.

| Generations | Illustrations | Location within the Plot | Orientation |
|---|---|---|---|
| g1 |  | edge, interior | east, south, west, north, center |
| g3 |  | edge | south, northeast |
| g5 |  | edge, interior | southeast, north |
| g7 |  | edge | southeast, east |
| g9 |  | edge | southeast, east |
| g11 |  | edge | southeast, east |
| g13 |  | edge, interior | southeast, center |

**Table 3.** *Cont.*

| Generations | Illustrations | Location within the Plot | Orientation |
|---|---|---|---|
| g15 |  | edge, interior | southeast, center, southwest |
| g17 |  | edge | southeast, east |
| g21 |  | edge | southwest, east |
| g23 |  | edge | southeast, east |
| g25 |  | edge | southeast |
| g27 |  | edge, interior | southwest, east |
| g29 |  | edge | southeast |
| g31 |  | edge | southeast |
| g39 |  | edge, interior | southeast, southwest, center |

**Table 4.** Statistics of distribution of larger squares.

| Item | Description | Percentage |
|---|---|---|
| Location | edge | 26% |
| | interior | 0% |
| | edge and interior | 74% |
| Orientation | north | 2.6% |
| | south | 5.2% |
| | east | 18.4% |
| | west | 0% |
| | southeast | 34.2% |
| | southwest | 10.5% |
| | northeast | 2.6% |
| | northwest | 0% |
| | center | 10.5% |

The last generation of solutions needed to be traded off before analyzing the typical solutions, and since the total sample of results was large due to the 55 Pareto solutions of the last generation, the ranking method [36] was adopted in this paper to reduce the complexity of the analysis by ranking the samples. In order to rank the used results, the following formula was adopted:

$$Rating = \frac{SU_{max} - SU_i}{SU_{max} - SU_{min}} + \frac{WU_{max} - WU_i}{WU_{max} - WU_{min}} + \frac{HP_{max} - HP_i}{HP_{max} - HP_{min}} \tag{3}$$

Note: $SU_i$ is the mean value of the summer UTCI, $SU_{max}$ is the maximum value of the summer UTCI, $SU_{min}$ is the minimum value of the summer UTCI, $WU_i$ is the mean value of the winter UTCI, $WU_{max}$ is the maximum value of the winter UTCI, $WU_{min}$ is the minimum value of the winter UTCI, $HP_i$ is the number of public spaces with sunshine hours less than 2 h, $HP_{max}$ is the maximum number of hours of sunshine in public spaces with sunshine hours less than 2 h and $HP_{min}$ is the minimum number of hours of sunshine in public spaces with sunshine hours less than 2 h.

This formula was used to find an equally weighted rating of the three objectives, and it could give different weights to each objective so that they could be ranked again according to the final score. After calculating the rating of solution by using the formula, the top ten solutions were selected as shown in Table 5.

**Table 5.** Weighted average of the top 10 solutions.

| Number | Winter UTCI Score | Summer UTCI Score | Sunshine Hours Score | Rating |
|--------|-------------------|-------------------|----------------------|--------|
| 1 | 0.67 | 0.38 | 0.85 | 1.9 |
| 2 | 0.54 | 0.59 | 0.76 | 1.89 |
| 3 | 0.62 | 0.4 | 0.84 | 1.86 |
| 4 | 0.71 | 0.46 | 0.67 | 1.85 |
| 5 | 0.63 | 0.68 | 0.52 | 1.83 |
| 6 | 0.63 | 0.69 | 0.52 | 1.83 |
| 7 | 0.62 | 0.69 | 0.52 | 1.83 |
| 8 | 0.8 | 0.21 | 0.81 | 1.83 |
| 9 | 0.46 | 0.51 | 0.85 | 1.82 |
| 10 | 0.52 | 0.41 | 0.87 | 1.81 |

From the table, it can be seen that the top ten solutions prioritized the three optimization objectives differently. The difference between the maximum and minimum scores of the winter UTCI was 0.28, which is not a big difference; the difference between the maximum and minimum scores of the summer UTCI was 0.48, which was somewhat large; and the difference between the maximum and minimum scores of the sunshine duration was 0.35, which is a small difference, so it can be deduced that the influence of the layout of the public spaces of the settlement on the three optimization objectives was ranked as follows: summer UTCI > sunshine duration > winter UTCI. Each solution in the list had different scores, among which the highest score for sunshine duration was 0.87 for solution 10, which had low UTCI scores for both the summer and winter, and solution 8 had the highest score for the winter UTCI, but this solution resulted in the lowest UTCI score in the summer and its total score was pulled down. The highest UTCI scores were found with solution 6 and solution 7, and only the winter UTCI scores were different between these two solutions, while the total scores and insolation were the same. In this paper, the top five solutions in the table were selected as typical cases for analysis, and these five cases were also balanced and met the three optimization objectives.

Firstly, the location of the public spaces in all five cases was concentrated in the center and southeast of the plot (Table 6), which had some reference value for planning the layout of the multistory residential areas considering the thermal environment in the summer and winter. Secondly, if we hope to have a more balanced optimization effect in the summer

and winter, the decentralized public space layout is better than the centralized one. Finally, if we hope to reduce the mean summer UTCI value, then the layout of a larger area of public spaces in the southeast location of a multistory settlement is favorable in terms of the summer wind penetration.

**Table 6.** Typical cases.

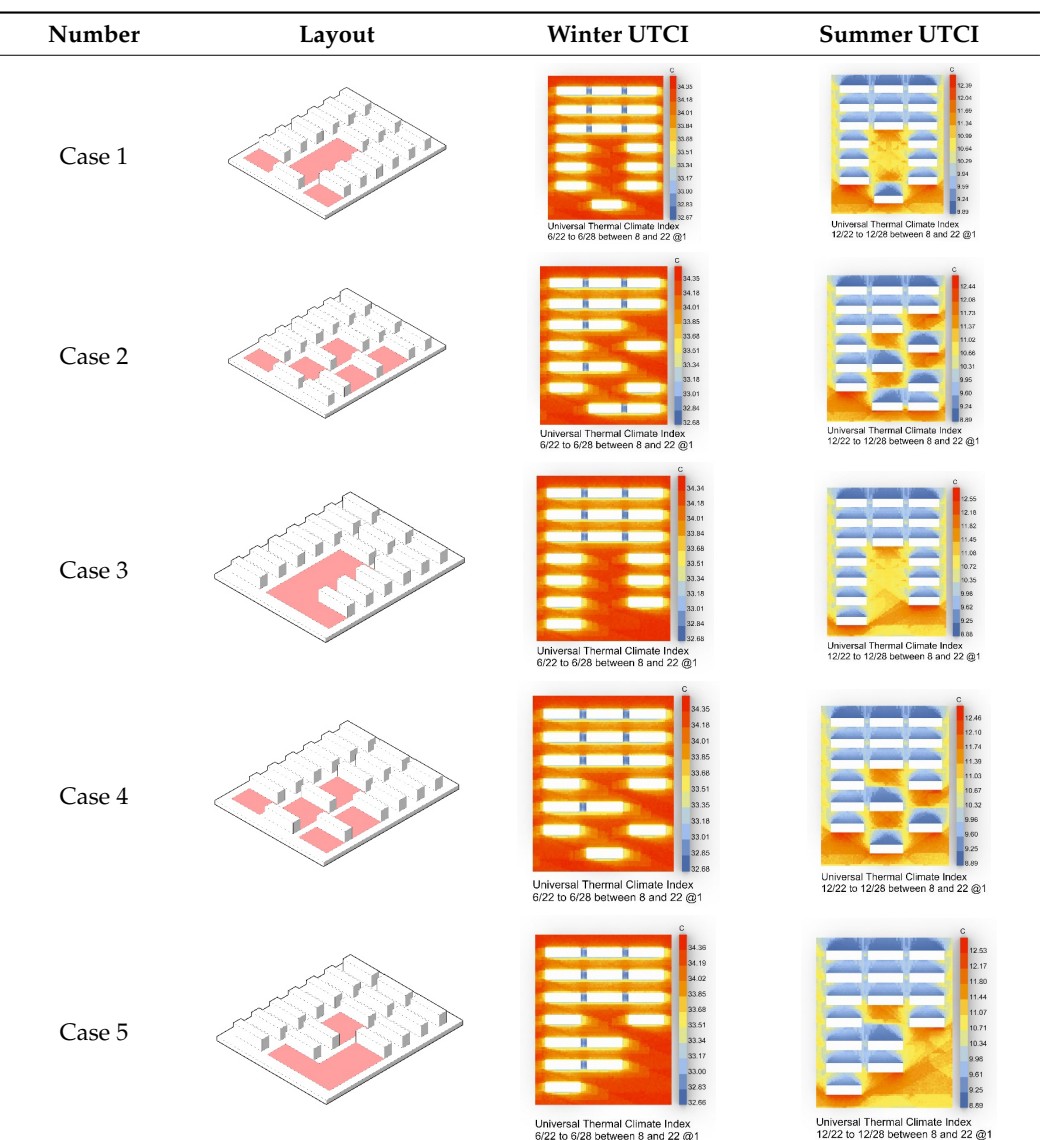

| Number | Layout | Winter UTCI | Summer UTCI |
| --- | --- | --- | --- |
| Case 1 | | | |
| Case 2 | | | |
| Case 3 | | | |
| Case 4 | | | |
| Case 5 | | | |

## 4. Discussion and Conclusions

### 4.1. Main Findings of This Study

After 39 generations of optimization, 55 valid solutions were obtained, in which the mean value of the summer UTCI decreased from 33.97 °C to 33.26 °C with a difference of 0.71 °C, and the mean value of the winter UTCI increased from 9.95 °C to 10.58 °C with a total increase of 0.63 °C. The ranking method was used to rank all the solutions in the last generation, and we selected the top five solutions as typical cases. An analysis was carried out, and it provided a programmatic reference for the layout design of multistory settlements in cold winter and hot summer areas.

Studies have demonstrated that strategic modifications to public space design hold immense potential in ameliorating the surrounding outdoor microclimate environment. In order to promote ventilation throughout various localities within the residential area, open nodes should be dispersed evenly while taking into consideration the need for optimal heat



retention practices in the winter and efficient cooling mechanisms in the summer. The wind velocity can be augmented by expanding the width and surface area of summer-facing openings, which in turn facilitates steady inlets of fresh air and promotes the air current flow throughout the residential area. Public spaces such as plazas should ideally be located in south-facing regions, while a layout pattern that gradually reduces the area of public spaces from the southeast to the northwest is conducive to mitigating the prevalence of chilly winter winds, which thus ensures a suitably warm and cozy living environment for residents.

*4.2. Public Space Design Strategy*

Based on the above experiments and analysis, the following design strategies are proposed for the layout [61] of public spaces in multistory settlements in hot-summer and cold-winter areas:

(1)    Consider the layout of the windward location in summer and winter

The main wind direction in Changsha in the winter is northwest, so reducing the amount of public spaces in the west and north side can effectively reduce the influence of the prevailing winds in the winter. Public spaces should be prevalent in the east and south side, which is conducive to the prevailing wind blowing in the summer and promotes the airflow in the internal passage. However, we should avoid creating vertical public spaces in areas with prevailing wind as this may produce a static wind effect, so the overall layout should include vertical public spaces in the north and west side as this will prevent the static wind effect. Therefore, the overall layout should include buildings on the north and west side, and the amount of public spaces in the east and south side should increase [62].

(2)    Uniform dispersion is better than dispersion

Uniform public spaces result in more shading being provided to the public spaces by buildings, which reduces the impact of solar radiation on the overall microclimate, while uniform and continuous public spaces are conducive to introducing the inflowing summer breeze into each public space, which has a certain effect on improving the local environmental comfort.

(3)    Larger squares are arranged in the south or southeast

Larger squares arranged in the south can help reduce the shading of sunlight [63,64] generated by buildings, while large south-facing squares can better introduce the summer southeast wind and improve the overall environmental comfort.

This paper analyzed the influence of public space layout elements on the thermal environment of multistory settlements in hot-summer and cold-winter regions. Due to the complexity of settlement forms and the differences between different climatic zones, the research results are only relevant to the design of settlements under specific conditions. In future research, the optimization of more types of settlements in different climate zones can be considered.

*4.3. Limitations and Future Work*

This study has some limitations that need to be addressed in future studies.

First, with regard to the scope of this study, this article exclusively scrutinized multi-layer residential areas, and although the typology of residential neighborhoods is diverse, research on high-rise residential areas and mixed-use residential areas is scarce. In upcoming research dedicated to the layout of public spaces within residential areas, high-rise residential areas or mixed-use residential areas ought to be the subject of scrutiny and further optimization through simulation.

Second, in terms of technical software, this article employed the LB toolset and octopus plugin embedded within GH for the simulation. When executing simulations of the thermal environment, Open studio was leveraged. However, if a Python program is developed, Envi-met could be considered to enhance the accuracy of the calculations.

Third, in relation to the design variables, the present study only focused on a few independent variables, namely the number, location and area of public spaces. Nonetheless, open-air public spaces are influenced by intricate environmental factors such as underlayment materials and spatial heights, which warrant further investigation.

Fourth, in the context of the optimization objectives, the geographical locale of interest was a region with a hot summer and cold winter, and summer and winter thermal comfort served as the optimization objective. If the region is a colder zone or a distinct climatic region, different optimization objectives must be selected in accordance with its respective climatic characteristics.

**Author Contributions:** Conceptualization, L.S.; methodology, Q.M.; software, M.C.; validation, S.L. and Q.M.; formal analysis, Q.M.; investigation, L.S.; resources, J.S.; data curation, Q.M.; writing—original draft preparation, Q.M.; writing—review and editing, Q.M.; visualization, F.Z.; supervision, L.S.; project administration, S.L.; funding acquisition, J.S. All authors have read and agreed to the published version of the manuscript.

**Funding:** This research was funded by Research on the Construction of Residential Open Space based on Health Needs (grant number: 21C0190) and Innovation Project for Postgraduates' Independent Exploration of Central South University (grant number: 2023ZZTS0264).

**Institutional Review Board Statement:** Not applicable.

**Informed Consent Statement:** Informed consent was obtained from all subjects involved in the study.

**Data Availability Statement:** The authors confirm that the data supporting the findings of this study are available within the article. The data are not publicly available due to privacy. Please contact corresponding author before use.

**Acknowledgments:** The authors of this paper are very grateful for the help provided by the following experts and people: Yaning An for the technical support and Fupeng Zhang for his valuable comments on the framework of the article.

**Conflicts of Interest:** The authors declare no conflict of interest.

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
