# Peer review of "Study on the Layout of Public Space in Multistory Settlements Based on Outdoor Thermal Environment in Hot-Summer and Cold-Winter Regions of China"

_atmosphere, doi:10.3390/atmos14071070_

Round 1

Reviewer 1 Report

This manuscript explores the optimization of residential public space in multi-story settlements in hot-summer and cold-winter areas using a parametric platform and optimization algorithm, resulting in improved urban microclimate and identified design strategies based on the Universal Thermal Climate Index.

Some comments that could improve the overall quality:

1. The abstract provides a brief overview of the study, but it could benefit from some improvements in terms of providing more context on the methodology, presenting specific results, and expanding on the derived design strategies.

2.  Overall, the introduction provides a clear background and rationale for the study, highlighting the gap in existing research and setting the stage for the investigation of performance-driven optimization for multi-story residential public space's layout in China's hot summer and cold winter regions.

3. Overall, I have no comments on the methodology.

4. However, I have comments on the result sections:

- The author mentions that two out of three dependent variables reached convergence after optimization iterations, leading to improvements in summer and winter UTCI values. However, the presence of fluctuations in the data during the optimization process raises questions about the stability and reliability of the obtained results. It is important to analyze the reasons for these fluctuations and assess their impact on the final conclusions.

- The author converted the three-dimensional coordinate system to a two-dimensional one due to having only two optimization objectives. While this simplification is understandable, it is crucial to ensure that the conversion accurately represents the relationship between the objectives and does not introduce any distortions or biases in the analysis.

- The analysis focuses on the relationship between mean UTCI values in summer and winter when analyzing the Pareto solutions. It is important to justify why these two objectives were chosen and whether they adequately represent the overall thermal comfort in the public spaces. Additionally, it would be helpful to discuss the trade-offs and compromises observed in the Pareto front and explain the implications for design decisions.

- The author conducted a statistical analysis of the positions of node space appearances during the optimization process. It is crucial to provide more details about the statistical methods used and the significance of the results. Without such information, it is difficult to evaluate the validity of the statistical analysis and its contribution to the overall findings.

5. Here are some comments for the conclusion section:
-In summary, while the conclusion provides some design strategies, it could be improved by addressing limitations, emphasizing the need for further validation, including quantitative assessment, and presenting the findings in a more organized and structured manner:

-The conclusion acknowledges the complexity of settlement forms and the differences between different climatic zones but does not explicitly address the limitations of the study. It would be beneficial to discuss any specific limitations, such as the generalizability of the findings to other contexts or potential confounding factors that were not considered.

-The conclusion states that the research results are only for reference in the design of settlements under specific conditions. However, it does not mention the need for further validation or replication of the findings in different settings or climates.

-The conclusion primarily presents qualitative design strategies without quantitative assessment or evaluation of their effectiveness. It would be valuable to provide some quantitative analysis or evidence to support the proposed strategies and demonstrate their potential impact on thermal comfort.

Here are my comments on the English/writing style:

1. Clarity and organization: Overall, the manuscript could benefit from better organization and clearer transitions between different ideas. Consider breaking it down into smaller paragraphs that focus on specific aspects, such as the shift from field measurements to computer simulations, the use of genetic optimization algorithms, and the software and platforms utilized. Each paragraph can then provide a more focused and coherent explanation of the respective topic.

2. Use of terminology and abbreviations: There are several abbreviations used throughout the text (e.g., OHM, PB, ST) without providing their full meaning or explanation. It's important to introduce and define abbreviations before using them consistently in the text to avoid confusion for readers.

3. The writing style often jumps between various ideas, studies, or findings without clear transitions or connections. Consider reorganizing the information to provide a more logical flow and better connect the different studies being referenced. This will make it easier for readers to understand the progression of research in the field.

4. Some sentences could benefit from restructuring or rephrasing to enhance clarity. Additionally, there are instances where word choice could be improved to convey ideas more precisely.

5. Grammar and sentence structure: There are a few instances where the sentences could be rephrased for clarity and improved readability. This warrants another round of proofreading.

6. Avoidance of repetition: The section repeats some information, such as the mention of genetic optimization algorithms and the software platforms (Rhino, Grasshoppers, Ladybug, etc.), in multiple sentences. Try to consolidate information and avoid unnecessary repetition to maintain conciseness.

Author Response

Point1 The abstract provides a brief overview of the study, but it could benefit from some improvements in terms of providing more context on the methodology, presenting specific results, and expanding on the derived design strategies.

Response1 I have added context on the methodology, presenting specific results, and expanding on the derived design strategies in abstract.

Point2 The author mentions that two out of three dependent variables reached convergence after optimization iterations, leading to improvements in summer and winter UTCI values. However, the presence of fluctuations in the data during the optimization process raises questions about the stability and reliability of the obtained results. It is important to analyze the reasons for these fluctuations and assess their impact on the final conclusions.

Response2 Additional explanation of fluctuations is provided at the end of section 3.1. 

Point3 The author converted the three-dimensional coordinate system to a two-dimensional one due to having only two optimization objectives. While this simplification is understandable, it is crucial to ensure that the conversion accurately represents the relationship between the objectives and does not introduce any distortions or biases in the analysis.

Response3 The reasons for the conversion of 3D coordinates to 2D coordinates are indeed unclear and have been added to the summary in 3.1.

Point4 The analysis focuses on the relationship between mean UTCI values in summer and winter when analyzing the Pareto solutions. It is important to justify why these two objectives were chosen and whether they adequately represent the overall thermal comfort in the public spaces. Additionally, it would be helpful to discuss the trade-offs and compromises observed in the Pareto front and explain the implications for design decisions.

Response4 The question of whether the UTCI is representative of the overall microclimate of the settlement has been added in 2.4.1. simulation of Performance.

Point5 The conclusion states that the research results are only for reference in the design of settlements under specific conditions. However, it does not mention the need for further validation or replication of the findings in different settings or climates.

Response5 Supplemented 4.3 Limitations and future work by writing the relevant content in this section.

Point6 The conclusion primarily presents qualitative design strategies without quantitative assessment or evaluation of their effectiveness. It would be valuable to provide some quantitative analysis or evidence to support the proposed strategies and demonstrate their potential impact on thermal comfort.

Response6 Added 4.1 Main findings of this study to add the specific data from the optimisation experiment to this section.

Point7 Clarity and organization: Overall, the manuscript could benefit from better organization and clearer transitions between different ideas. Consider breaking it down into smaller paragraphs that focus on specific aspects, such as the shift from field measurements to computer simulations, the use of genetic optimization algorithms, and the software and platforms utilized. Each paragraph can then provide a more focused and coherent explanation of the respective topic.

Response7 Subsections 2.2 and 2.3 have been more carefully divided and more precisely named.

Point8 Use of terminology and abbreviations: There are several abbreviations used throughout the text (e.g., OHM, PB, ST) without providing their full meaning or explanation. It's important to introduce and define abbreviations before using them consistently in the text to avoid confusion for readers.

Response8 An explanation of the abbreviations has been added in the appropriate place.

Point9 The writing style often jumps between various ideas, studies, or findings without clear transitions or connections. Consider reorganizing the information to provide a more logical flow and better connect the different studies being referenced. This will make it easier for readers to understand the progression of research in the field.

Response9 The writing style of the literature review section has been adapted.

Point10 Some sentences could benefit from restructuring or rephrasing to enhance clarity. Additionally, there are instances where word choice could be improved to convey ideas more precisely.

Response10 A number of inappropriate uses of words and phrases in the text have been revised.

Point11 Grammar and sentence structure: There are a few instances where the sentences could be rephrased for clarity and improved readability. This warrants another round of proofreading.

Response11 Grammar and sentence structure was checked and revised.

Point12 Avoidance of repetition: The section repeats some information, such as the mention of genetic optimization algorithms and the software platforms (Rhino, Grasshoppers, Ladybug, etc.), in multiple sentences. Try to consolidate information and avoid unnecessary repetition to maintain conciseness.

Response12 Repeated descriptions of software and theory replaced with abbreviations.

Reviewer 2 Report

The article examines the design strategies for public spaces in multi-storey settlements located in hot-summer and cold-winter regions. There are a few suggestions to further enhance its clarity and coherence

1.Consider organizing the overall optimization results in subsections within the results section. This approach will help readers navigate the findings more effectively and understand the outcomes in a structured manner.

2.In the conclusion section, it is recommended to avoid repetition of the first and second points. Instead, focus on refining and summarizing the main findings and their implications. This will strengthen the concluding remarks and avoid redundancy.

3. For result 3.3, it is suggested to supplement the relationship between windward opening and UTCI (Universal Thermal Climate Index). While the table provides valuable information, it may not fully capture the impact of upwind opening rates on UTCI. This addition will strengthen the findings and enrich the discussion surrounding the optimization of multi-storey settlements in hot-summer and cold-winter regions.

4. The definition of public space in multi-storey settlements is missing in the article, and the reader is not clear about the object of the study, so it is suggested to add the definition of concepts related to multi-storey settlements.

5. the tables in 3.4. larger squares do not correspond to the text and should be supplemented with the corresponding data analysis.

Moderate editing of English language

Author Response

Point1 Consider organizing the overall optimization results in subsections within the results section. This approach will help readers navigate the findings more effectively and understand the outcomes in a structured manner.

Response1 The overall results have been included in a separate subsection.

Point2 In the conclusion section, it is recommended to avoid repetition of the first and second points. Instead, focus on refining and summarizing the main findings and their implications. This will strengthen the concluding remarks and avoid redundancy.

Response2 The duplicate conclusion has been removed and the conclusion sheet has been added.

Ponit3  For result 3.3, it is suggested to supplement the relationship between windward opening and UTCI (Universal Thermal Climate Index). While the table provides valuable information, it may not fully capture the impact of upwind opening rates on UTCI. This addition will strengthen the findings and enrich the discussion surrounding the optimization of multi-storey settlements in hot-summer and cold-winter regions.

Response3 Relevant information has been added to section 3.3

Point4 The definition of public space in multi-storey settlements is missing in the article, and the reader is not clear about the object of the study, so it is suggested to add the definition of concepts related to multi-storey settlements.

Response4 Definitions are added in the introduction.

Point5 The tables in 3.4. larger squares do not correspond to the text and should be supplemented with the corresponding data analysis.

Response5 Table 4 has been supplemented in subsection 3.5 for statistical location distributions.

Reviewer 3 Report

This study optimized the outdoor space of a residential area by adopting parametric study. The topic is relatively new, but it suffers from writing problems. The following comments can be considered to improve the quality of manuscript:

1. "Performance-driven" is too general, directly indicate in the title what has been optimized. E.g. outdoor thermal environment or outdoor thermal comfort. 

2. The overall writing needs significant improvement. For example, line 20, development of urbanization is strange, because urbanization itself already indicate the process. Line 39, Pennsylvania professor should be professor from University of Pennsylvania . Lines 136 to 149, these sentences should be capitalized. Too many writing errors that needs to be addressed.

3. When cite an author, no need to indicate authors' afflications and first name.

4. There have been studies on using optimization to search for designs that led to better thermal comfort.  You should cite them, for example: 

Sun, R., Liu, J., Lai, D. and Liu, W., 2023. Building form and outdoor thermal comfort: Inverse design the microclimate of outdoor space for a kindergarten. Energy and Buildings284, p.112824.

5. Chinese in many figures, please check.

6. What is your time of study? Is it for an hour, a day or a longer duration? I cannot find the information. 

7. Reference 16 and 48 are the same.

8. Caption of Figure 15 is not correct. Should be sunshine duration. 

Should be much improved.

Author Response

Dear reviewer,

Thank you so much for the comments, here's my revision notes:

Point1 "Performance-driven" is too general, directly indicate in the title what has been optimized. E.g. outdoor thermal environment or outdoor thermal comfort. 

Response1 As suggested, the title has been changed

Point2 The overall writing needs significant improvement. For example, line 20, development of urbanization is strange, because urbanization itself already indicate the process. Line 39, Pennsylvania professor should be professor from University of Pennsylvania . Lines 136 to 149, these sentences should be capitalized. Too many writing errors that needs to be addressed.

Response2  line 20, modifications have been made and it is now line24. Line 39, Pennsylvania professor has been replaced with professor from University of Pennsylvania . Lines 136 to 149, these sentences has been modified to capitalized.

Point3 When cite an author, no need to indicate authors' afflications and first name.

Response3 The surname and institution of the author of the citation have been removed

Point4 There have been studies on using optimization to search for designs that led to better thermal comfort.  You should cite them, for example: 

Sun, R., Liu, J., Lai, D. and Liu, W., 2023. Building form and outdoor thermal comfort: Inverse design the microclimate of outdoor space for a kindergarten. Energy and Buildings284, p.112824.

Response4 I've added this article to reference 16

Point5 Chinese in many figures, please check.

Response5 I checked and replaced Figure 8 and Figure 17.

Point6 What is your time of study? Is it for an hour, a day or a longer duration? I cannot find the information. 

Response6 Time added at lines 321 and 448.

Point7 Reference 16 and 48 are the same.

Response7 I have replaced reference 16 with the one you recommended.

Point8 Caption of Figure 15 is not correct. Should be sunshine duration. 

Response8 The caption of Figure 15 has been changed to sunshine duration. 

Round 2

Reviewer 3 Report

The paper has been revised according to my comments, it can be accepted.